# One-Step Synthesis of Cu_x_O_y_/TiO_2_ Photocatalysts by Laser Pyrolysis for Selective Ethylene Production from Propionic Acid Degradation

**DOI:** 10.3390/nano13050792

**Published:** 2023-02-21

**Authors:** Juliette Karpiel, Pierre Lonchambon, Frédéric Dappozze, Ileana Florea, Diana Dragoe, Chantal Guillard, Nathalie Herlin-Boime

**Affiliations:** 1NIMBE, CEA, CNRS, Université Paris-Saclay, CEA Saclay, 91191 Gif-sur-Yvette, France; 2Institut de Recherche Sur La Catalyse Et l’Environnement De Lyon (IRCELYON), Université Lyon 1, CNRS, Avenue Albert Einstein, 69626 Villeurbanne, France; 3Laboratory of Physics of Interfaces and Thin Films (LPICM), Ecole Polytechnique, CNRS, Institut Polytechnique de Paris, 91128 Palaiseau, France; 4CNRS, Institut de Chimie Moléculaire et des Matériaux d’Orsay (ICMMO), Université Paris-Saclay, 91405 Orsay, France

**Keywords:** photocatalysis, ethylene, hydrocarbons, hydrogen, titanium dioxide, propionic acid

## Abstract

In an effort to produce alkenes in an energy-saving way, this study presents for the first time a photocatalytic process that allows for the obtention of ethylene with high selectivity from propionic acid (PA) degradation. To this end, TiO_2_ nanoparticles (NPs) modified with copper oxides (Cu_x_O_y_/TiO_2_) were synthetised via laser pyrolysis. The atmosphere of synthesis (He or Ar) strongly affects the morphology of photocatalysts and therefore their selectivity towards hydrocarbons (C_2_H_4_, C_2_H_6_, C_4_H_10_) and H_2_ products. Specifically, Cu_x_O_y_/TiO_2_ elaborated under He environment presents highly dispersed copper species and favours the production of C_2_H_6_ and H_2_. On the contrary, Cu_x_O_y_/TiO_2_ synthetised under Ar involves copper oxides organised into distinct NPs of ~2 nm diameter and promotes C_2_H_4_ as the major hydrocarbon product, with selectivity, i.e., C_2_H_4_/CO_2_ as high as 85% versus 1% obtained with pure TiO_2_.

## 1. Introduction

Ethylene (C_2_H_4_) is a key molecule in the chemical industry. About 75% of petrochemical products are obtained from this light olefin, including intermediates such as ethylene glycol and polymers such as polyethylene or polyvinyl chloride [1,2,3]. Currently, ethylene is mostly produced from the steam cracking of fossil fuels. However, besides relying on non-sustainable feedstock and being extremely polluting, such a method is one of the most energy-consuming processes in the chemical sector as it requires considerable working temperatures (T > 750 °C) [4,5].

Hence, researchers are developing new strategies for cleaner and lower-cost ethylene production without the use of hydrocarbons. Among them, heterogeneous photocatalysis attracts great attention for being an oxidation process operating at ambient temperature and atmospheric pressure, thus offering favourable perspectives for sustainable organic synthesis using solar light [6]. Nonetheless, very few research articles describe the photocatalytic production of ethylene [7,8,9,10,11,12]. Despite its high relevance, CO_2_ photocatalytic reduction to C_2_H_4_ is very challenging as it requires effective C-C coupling and a multi-electron reaction process [7]. Another strategy to form ethylene consists of the photocatalysis of carboxylic acids operating in an anaerobic medium. Photo-decarboxylation of such acids (acetic, propionic, butanoic, etc.) mainly produces CO_2_, H_2_, and alkanes [8,9,10,11,12,13] using TiO_2_ or TiO_2_-based nanomaterials decorated with metal co-catalysts to promote quantum yield maximization [14]. In some cases, alkenes were also detected. Kraeutler and Bard [11] identified ethylene traces from propionic acid decarboxylation with Pt/TiO_2,_ which were confirmed afterwards in the Betts et al. study using TiO_2_ Evonik P25 [12]. Scandura et al. enhanced propene and ethylene selectivities from butyric acid and propionic acid photoreforming, respectively, with Au/TiO_2_ compared to Pt/TiO_2_ [10]. However, despite the use of onerous noble metals, alkenes were each time detected in trace amounts.

Herein, we report for the first time the selective photocatalytic production of ethylene from the valorisation of propionic acid using TiO_2_ modified with copper nanoparticles under UVA light. Specifically, TiO_2_ and Cu_x_O_y_/TiO_2_ photocatalysts were elaborated using a laser pyrolysis technique, an industrial-scale process that enables a one-step and continuous synthesis of nanopowders at several grams per hour. The impact of the laser pyrolysis operating atmosphere (Ar, He) on the morphology and the material photocatalytic performances were studied. Propionic acid (PA) was selected as the model molecule; this volatile fatty acid can be obtained from the fermentation of organic wastes [15]. This work provides a new method for sustainable ethylene production with low energy consumption and low-cost photocatalysts.

## 2. Materials and Methods

### 2.1. Chemicals

Titanium (IV) isopropoxide (TTIP, ≥97% purity), ethylacetate (99.8% purity) and copper (II) acetylacetonate (Cu(acac)_2_, ≥97% purity) were purchased from Sigma-Aldrich, Darmstadt, Germany. Pure o-xylene was supplied from Fisher, Waltham, MA, USA. Helium He, argon Ar and ethylene C_2_H_4_ gases were supplied by Air Products and Chemicals, Allentown, PA, USA. All the reactants were used without further purification. Propionic acid (PA) was obtained from Sigma-Aldrich, Darmstadt, Germany (≥99.5% purity).

### 2.2. Chemicals Synthesis of TiO_2_ and Cu_x_O_y_/TiO_2_ Nanoparticles

The synthesis of TiO_2_ and Cu modified TiO_2_ photocatalysts was performed with the laser pyrolysis technique under He or Ar gas. This method permits the continuous and direct production of nanopowders and is described elsewhere [16]. Briefly, its principle is based on the orthogonal interaction of an infrared CO_2_ laser (10.6 μm) in continuous mode with a gaseous or liquid mixture of reagents. The laser was focused by means of a cylindrical lens, resulting in a horizontal beam of about 30 mm. For the synthesis of reference samples (TiO_2_), the mixture consisted of 175 g of liquid titanium (IV) isopropoxide (TTIP) dissolved in a 150 mL o-xylene/ethylacetate solution (65 vol%/35 vol%). Dissolution of 4.13 g of Cu(acac)_2_ (copper (II) acetylacetonate) into the previous mixture permitted us to obtain Cu_x_O_y_/TiO_2_ particles with a theoretical Cu content equalling 2.0 wt% regarding the final nanomaterials. The resulting liquid was converted into microdroplets with a spray generator, regulating the mixture at 30 °C. The liquid droplets were transported into the reactor zone through a carrier gas (He or Ar) at 2000 cm^3^.min^−1^ flow. A sensitiser gas (C_2_H_4_, 800 cm^3^.min^−1^) was added to the carrier gas to enhance the laser absorption of the precursors. The reaction zone was confined by He or Ar inert gas flow. The CO_2_ laser power (measured under inert gas) was set to 670 W and the pressure inside the reactor was regulated to 740 Torr. The powders were collected on filters located downstream.

The TiO_2_ and Cu_x_O_y_/TiO_2_ as-formed materials had the appearance of black, grey or brown-coloured powders due to the presence of carbon impurities. All samples were further annealed under air at 450 °C during 6 h in a tubular furnace (Carbolite) to remove residual carbon. The annealed particles were labelled TiO_2_-X and Cu/TiO_2_-X (X = He, Ar) depending on their synthesis environment. Powder images are provided in Appendix A.

### 2.3. Samples Characterization

The morphology of prepared photocatalysts was evaluated using transmission electron microscopy (TEM, Tecnai G2 F20 from Fei, Hillsboro, OR, USA).

In order to probe the presence of Cu element, energy-dispersive spectroscopy (EDS) analyses were performed in the scanning transmission electron microscope high-angle annular dark field (STEM-HAADF) imaging mode of a 200 kV Titan-Themis TEM/STEM electron microscope from Fei (Hillsboro, OR, USA) equipped with a Cs probe corrector and a ChemiSTEM Super-X detector (Fei, Hillsboro, OR, USA). These two accessories allow chemical mapping of light and heavy elements with a spatial resolution in the picometer range. The experimental conditions were set so that the total current within the probe used for the STEM-HAADF EDS chemical analysis was about 85 pA.

Samples specific surface areas were determined using N_2_ adsorption according to the Brunauer–Emmett–Teller (BET) method with a Micromeritics (Norcross, GA, USA) 3Flex apparatus. Briefly, the BET method is based on the measurement of N_2_ gas molecule physisorption onto the solid sample.

Crystalline phases were identified via X-ray diffraction (XRD) measurements using a Bruker D2 Phaser diffractometer (Bruker AXS, Karlsruhe, Germany) with monochromatised Cu Kα radiation (λ = 1.5418 Å). Anatase (JCPDS No. 21-1272) and rutile (JCPDS No. 21-1276) percentages were calculated from the Spurr and Myers equation [17].

XPS measurements were performed on a K-Alpha spectrometer from ThermoFisher, Waltham, MA, USA equipped with a monochromated X-ray source (Al Kα, 1486.7 eV) with a spot size of 400 µm. The hemispherical analyser was operated in CAE (constant analyser energy) mode, with a pass energy of 200 eV and a step of 1 eV for the acquisition of survey spectra and a pass energy of 50 eV and 10 eV and a step of 0.1 eV for the acquisition of narrow scans. The spectra were recorded and treated by means of Avantage software (version 5.967). The binding energy scale was calibrated against the Ti 2p binding energy set at 258.5 eV.

Copper content was measured using inductively coupled plasma optical emission spectroscopy (ICP-OES) with a Jobin Yvon Activa instrument (Horiba, Edison, NJ, USA).

Carbon content was determined by using a Emia-320V Carbon/Sulphur analyser (Horiba, Edison, NJ, USA).

### 2.4. Photocatalytic Measurements

Photocatalytic runs were carried out in a 250 mL Pyrex photoreactor equipped with a mechanic agitator (500 rpm) and a gas inlet and outlet for gas sampling and purge. A 18W PL-L lamp (Philips, Amsterdam, The Netherlands), placed below the photoreactor, was chosen as the UVA source with a UV band of 350-410 nm centred at 370 nm (irradiance of 5 mW.cm^−2^).

We introduced 100 mL of a 1.0 vol% PA solution containing 50 mg of photocatalyst into the reactor. A continuous flow of argon (70 mL.min^−1^) was applied through a bubbler for 6 h into the system to fully remove the ambient air. After the purge, the photoreactor was sealed and irradiated with UVA light. Every 65 min, a 2 mL aliquot of photoproduced gases in the headspace was carried by vacuum pumping into a Clarus 500 GC FID-PDHID (PerkinElmer, Waltham, MA, USA) for analysis. Products were firstly separated though a RT-Q-Bond (Restek–30 m × 0.53 mm × 20 μm) column before FID with a Polyarc methanization module. For PDHID, a RT-M5A molecular sieve (Restek–30 m × 0.32 mm × 30 μm) was added. The carrier gas was helium N60 grade (Messer, Bad Soden, Germany). For each gaseous analysis, the oven temperature was set at 50 °C rising at 20 °C/min to reach a maximum temperature of 150 °C maintained for an additional 55 min. Tests were repeated twice to ensure reproducibility.

## 3. Results and Discussion

### 3.1. Physico-Chemical Properties

#### 3.1.1. Synthesis

It was possible to observe the flame during synthesis experiments through a window. The flame seen in presence of helium was much less bright than under argon, which is an indication of a lower flame temperature under He atmosphere. The temperature appears to be a major parameter due to the change of atmosphere. The specific heat capacity of helium is 10 times higher than argon, explaining why such a lower temperature was reached in helium atmosphere after laser absorption as well as the fast temperature decrease. As a consequence, the lower temperature observed under He atmosphere explains a lower grain size as well as a lower rutile content in nanoparticles synthesised under He compared to nanoparticles generated under Ar (Table 1 and Figure 1). Similar observations were reported by Pignon et al. [16], who explained these results as a cooling effect of He as well as less efficient confinement of the reactants. Indeed, He is a light gas and is therefore less efficient to confine the reaction. This is also apparent from the lower production rate (i.e., collected powder, Table 1) in He compared to Ar.

Table 1 summarises the characteristics of the obtained Cu_x_O_y_/TiO_2_ samples and their TiO_2_ references synthesised in similar conditions. In agreement with the more efficient decomposition of precursors, the production rate is higher under Ar atmosphere and carbon content increases in the as-formed TiO_2_-Ar and Cu/TiO_2_-Ar samples compared to samples obtained under He environment.

After annealing, TiO_2_ samples turned white, indicating the efficient removal of carbon species, whereas Cu_x_O_y_/TiO_2_ powders appeared pale green-coloured, suggesting the presence of copper. The associated mass losses are in agreement with the carbon content of the as-formed powders. This result was confirmed with elementary analyses revealing only traces of residual carbon ≤0.3 wt% in all the samples.

#### 3.1.2. Morphology and Structure

The morphology of the annealed samples was first investigated by TEM analysis (Figure 1 and Appendix A). From the TEM images, all particles appear to present mostly a spherical shape arranged in chains typical of gas phase synthesis. The average diameters estimated from both BET and TEM measurements are reported in Table 1 and correspond well to each other.

The grain size is larger for nanoparticles synthetised under Ar rather than He environment. Particularly, the diameter of Cu/TiO_2_-Ar nanoparticles deduced from the TEM analyses seems to be 1.9 times larger than its bare TiO_2_-Ar reference. It seems that the addition of copper element promotes the formation of larger particles. This observation together with the highest rutile content observed in this sample confirms a higher synthesis temperature for Cu/TiO_2_-Ar. XRD analyses were performed on all the samples before and after annealing (Appendix A). The samples consist of a mixture of a major anatase phase with a minor rutile phase, except for Cu/TiO_2_-Ar (Table 1). Finally, XRD patterns do not reveal peaks associated with copper species, probably due to small crystallite sizes and low Cu content.

UV-visible DRS optical analyses (Appendix A) revealed a band gap between 3.0 and 3.2 eV, in good agreement with anatase (3.2 eV) and rutile (3.0 eV) corresponding ratios in the annealed nanoparticles.

#### 3.1.3. Chemical Analyses

The presence of copper was first confirmed by ICP-OES analysis (Table 1). Cu loadings are consistent regarding the 2.0 wt% Cu content in the liquid precursors. Regarding the copper content, it is slightly higher, with 1.91 ± 0.04 wt% (Cu/TiO_2_-Ar) vs. 1.76 ± 0.04 wt% (Cu/TiO_2_-He) when the argon atmosphere was used. The reaction temperature appears higher when working under argon atmosphere compared to helium atmosphere. This higher temperature induces a better decomposition of copper precursor in argon atmosphere and therefore a slightly higher Cu content in the powder.

High-angle annular dark-field imaging (STEM-HAADF) coupled with EDS chemical analysis was employed in order to evidence the presence as well as the repartition of Ti and Cu elements within the analysed samples. From the STEM-HAADF images illustrated in Figure 2 and Appendix A, the presence of bright spots was detected on the surface of Cu/TiO_2_-Ar (Figure 2a,c). The corresponding STEM-HAADF EDS map recorded by considering the Cu Kα edge illustrated in Figure 2b suggests that these bright spots within the Cu/TiO_2_-Ar powders are Cu-based nanoparticles localised within the whole TiO_2_ support. From these images, we deduce that the size of these NPs is of 1.8 nm ± 0.3 nm (Appendix A). Moreover, STEM-HAADF EDS line scan analyses performed across the yellow line Cu/TiO_2_-Ar (Figure 2c,d) confirm the presence of Cu species into four distinct NPs. Moreover, these Cu NPs appear localised on the surface of TiO_2_. On the contrary, STEM-HAADF analyses performed on Cu/TiO_2_-He sample (Figure 2e,f) do not show any defined bright spot. The inset on the corresponding EDS-Cu elemental map associated with the Cu/TiO_2_-He image (Figure 2f) highlights a very poor contrast between Cu element (in black) and the background (in white). From this inset, it can be seen Cu well-dispersed species across TiO_2_ particles. The STEM-HAADF EDS line scan analysis performed along the yellow arrow (Figure 2g,h) confirms that Cu element is homogeneously dispersed within TiO_2_, most likely due to the lower specific surface area of TiO_2_ nanoparticles when synthetised under He. From this type of analysis, we can sustain that there are significant differences in terms of Cu dispersion and size within the TiO_2_ support depending on the material synthesis conditions, i.e., the atmosphere in the present case.

XPS analyses were conducted to investigate the surface chemical state of elements in both TiO_2_ and Cu_x_O_y_/TiO_2_ synthetised under different atmospheres. Appendix A depicts the XPS full spectra of pure TiO_2_ and Cu/TiO_2_, which shows the presence of, Ti, O, and C for all samples. In addition, Cu element appeared in both Cu/TiO_2_-He and Cu/TiO_2_-Ar powders. The Ti 2p core-level spectrum of TiO_2_ (Table 2 and Appendix A) consists of two peaks centred at 464.2 and 458.5 eV, corresponding to Ti 2p_1/2_ and Ti 2p_3/2_ of Ti^4+^, respectively [18]. A broadening of the Ti 2p doublet is observed for Cu_x_O_y_/TiO_2_ samples in comparison to TiO_2_ references (Table 2). Indeed, Ti 2p^3/2^ FWHM corresponds to 1.1 eV for both TiO_2_-He and TiO_2_-Ar and increases to 1.3 eV and 1.5 eV for the Cu/TiO_2_-He and Cu/TiO_2_-Ar samples, respectively. This suggests an interaction between copper species and TiO_2_ through the incorporation of Cu ions in TiO_2_ lattice [19,20].

The Cu 2p_3/2_ core-level spectrum (Figure 3) reveals several co-existing Cu chemical states in Cu/TiO_2_-He and Cu/TiO_2_-Ar [18,21]. The peaks at 934.8 eV and 933.2 eV are assigned to the Cu^2+^ 2p_3/2_ signal of Cu(OH)_2_ and CuO, respectively. Shake-up satellite peaks confirm the presence of Cu^2+^. Additionally, the component at about 932.5 eV refers to both the Cu^0^ and Cu^+^ 2p_3/2_ signal, as the two oxidation states cannot be distinguished, having very close binding energies. By comparing Cu^2+^ proportions, it seems that copper species are more reduced in the Cu/TiO_2_-He sample. XPS quantification reveals copper content of 4.1 wt% and 9.2 wt% for Cu/TiO_2_-He and Cu/TiO_2_-Ar, respectively. This is 2.3 and 4.9 times higher than bulk values provided by ICP-OES analyses, confirming a significant segregation of copper species on the surface of TiO_2_, clearly pronounced in the case of the Cu/TiO_2_-Ar sample. In addition, the Cu/Ti ratio is lower for Cu/TiO_2_-He, which could be attributed to a higher dispersion on the support [22,23], confirming our previous results.

Finally, the TiO_2_ and Cu_x_O_y_/TiO_2_ nanoparticles obtained by laser pyrolysis exhibit distinct morphologies: the Cu/TiO_2_-He sample consists of nanoparticles of 10 nm range, including very well dispersed Cu species on TiO_2_, whereas the Cu/TiO_2_-Ar particles are about 30 nm, with defined copper-based nanoparticles of about 2 nm size on the TiO_2_ support.

### 3.2. Photocatalytic PA Degradation in Anaerobic Media

The main gases photoproduced from PA degradation under anaerobic conditions are presented in Figure 4 and a typical chromatogram is shown in Appendix A. The identified gases are CO_2_, ethylene C_2_H_4_, ethane C_2_H_6_ and H_2_. Butane C_4_H_10_ (Appendix A) was detected as a minor product. The principal mechanisms for saturated carboxylic acid degradation in oxygen-free media with pure TiO_2_ or decorated with noble metals have already been reported [11,13]. The first step consists of the decarboxylation of carboxylic acid through reaction with photo-generated trapped holes (Equation (1)):CH_3_CH_2_COOH + h^+^→CH_3_CH_2_● + CO_2_ + H^+^,(1)

The CH_3_CH_2_● radical can react with a H● (H^+^ + e^−^) to further form C_2_H_6_ in majority. The coupling of two CH_3_CH_2_● is responsible for C_4_H_10_ formation, and two H● can form H_2_. According to Scandura et al. [10], traces of ethylene are formed by the reaction of CH_3_CH_2_● with h^+^. Hence, from Equation (1), selectivities are defined in terms of hydrocarbon (C_2_H_4_, C_2_H_6_) to CO_2_ ratio.

Several differences can be seen from Figure 4. With pristine TiO_2_ (black and blue curves, Figure 4), the main products are CO_2_ and C_2_H_6_ (C_2_H_6_/CO_2_ = 73% and 68% for TiO_2_-He and TiO_2_-Ar, respectively). The slightly different photoactivities of TiO_2_-He and TiO_2_-Ar observed for CO_2_ and C_2_H_6_ productions could be correlated with different anatase to rutile ratios (i.e., presence of junction) as well as different BET surface areas. It is known that the presence of a junction enhances activity through more efficient electron transfer; see, for example [24,25]. However, in the present case, it is difficult to conclude if the slightly better efficiency of TiO_2_-He is related to its largest surface area (111 vs. 81 m^2^.g^−1^) or to the change in anatase to rutile ratio (87 vs. 76% of anatase). H_2_ and C_2_H_4_ (C_2_H_4_/CO_2_ = 1%) are identified in trace amounts, in agreement with the literature in the absence of co-catalysts for H_2_ [26,27,28] or with the use of noble metals (Pt, Au) for C_2_H_4_ [10,11,12].

Surface modifications of TiO_2_ photocatalysts with copper/copper oxides leads to drastic changes in terms of levels of photo-generated products. Notably, Cu/TiO_2_-He appears as the most efficient photocatalyst for CO_2_, C_2_H_6_, and C_4_H_10_ production (Appendix A), with C_2_H_6_ as the major hydrocarbon (green curve, Figure 4). C_2_H_4_ formation is enhanced (C_2_H_4_/CO_2_ = 11%) until 200 min of reaction, then it slows down. On the contrary, H_2_ production is increased from this same irradiation time (200 min). These phenomena could be attributed to the efficiency of the Cu_x_O_y_-TiO_2_ heterojunction that decreases electron–hole recombination [29,30], but above all thanks to the reduction of copper oxides into metallic copper [31,32,33]. Indeed, well-dispersed copper species on the TiO_2_ surface in the Cu/TiO_2_-He sample could be reduced after about 200 min of UV irradiation. Therefore, Cu^0^_,_ acting as an e^-^ scavenger, highly enhances H_2_ formation. The CH_3_CH_2_● radical preferentially reacting with a H● formed on the Cu^0^ site could explain how reduced copper species inhibit the formation of C_2_H_4_ while enhancing C_2_H_6_ production.

Remarkably, Cu/TiO_2_-Ar (red curve, Figure 4) is the most efficient material for C_2_H_4_ production: it is the major hydrocarbon product with a C_2_H_4_/CO_2_ ratio in the range = 85-88%. Its formation evolves linearly at 260 ppmv/h, 130 times greater than the production rate reached from pure titania. This efficient production is obtained at the expense of C_2_H_6_ (C_2_H_6_/CO_2_ = 4–7%) and H_2_ formation. This contrasting behaviour compared to Cu/TiO_2_-He may be related to less-dispersed, larger, and oxidised copper nanoparticles on TiO_2_ support, as seen in STEM-HAADF/EDS and XPS characterizations. The larger size allows Cu_x_O_y_ NPs to resist complete photo-reduction, as reported by Imizcoz et al. [31]. In addition, according to Yu et al. [29], a larger Cu_x_O_y_ size limits the direct transfer of electrons from Cu_x_O_y_ to H^+^ due to a decrease in the conduction band. Consequently, H_2_ and C_2_H_6_ formations are not favoured. These particular conditions seem to enhance C_2_H_4_ production, as observed in the present study.

## 4. Conclusions

This study reports for the first time the significant production of C_2_H_4_ molecules from the photocatalytic degradation of propionic acid. TiO_2_ nanoparticles modified with Cu_x_O_y_ used as photocatalysts were prepared using the laser pyrolysis synthesis method. The atmosphere of synthesis (argon or helium) strongly modifies Cu_x_O_y_/TiO_2_ morphologies as well as their photocatalytic properties. Cu_x_O_y_/TiO_2_ obtained under helium consist of small TiO_2_ nanoparticles with highly dispersed copper species that could be reduced into Cu^0^ under UV irradiation. These reduced species promote the formation of H_2_ and C_2_H_6_ at the expense of C_2_H_4_. On the contrary, larger Cu_x_O_y_/TiO_2_ NPs were obtained under argon atmosphere with defined Cu_x_O_y_ nanoparticles (1–2 nm diameter) on the surface of TiO_2_. It seems that such less-dispersed Cu_x_O_y_ nanoparticles inhibit H_2_ and C_2_H_6_ formation. The major hydrocarbon was C_2_H_4_, with an outstanding selectivity higher than 85%, outperforming by 130 times the ethylene production from pristine TiO_2_. Although the ethylene formation mechanism should be further examined, this study paves the way to a new energy-saving method to produce C_2_H_4_ from organic acid without noble-metal-based photocatalysts.

## Figures and Tables

**Figure 1 nanomaterials-13-00792-f001:**
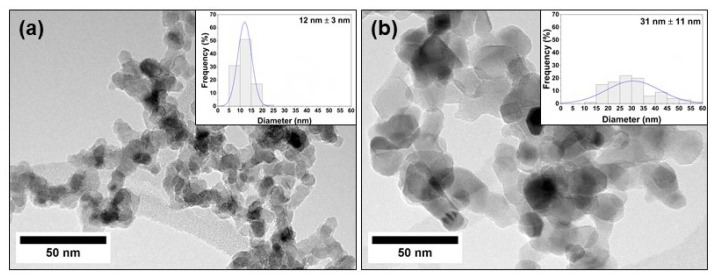
TEM images of annealed Cu/TiO_2_-He NPs (**a**) and Cu/TiO_2_-Ar NPs (**b**); insets refer to associated size distribution histogram fitted with the normal function.

**Figure 2 nanomaterials-13-00792-f002:**
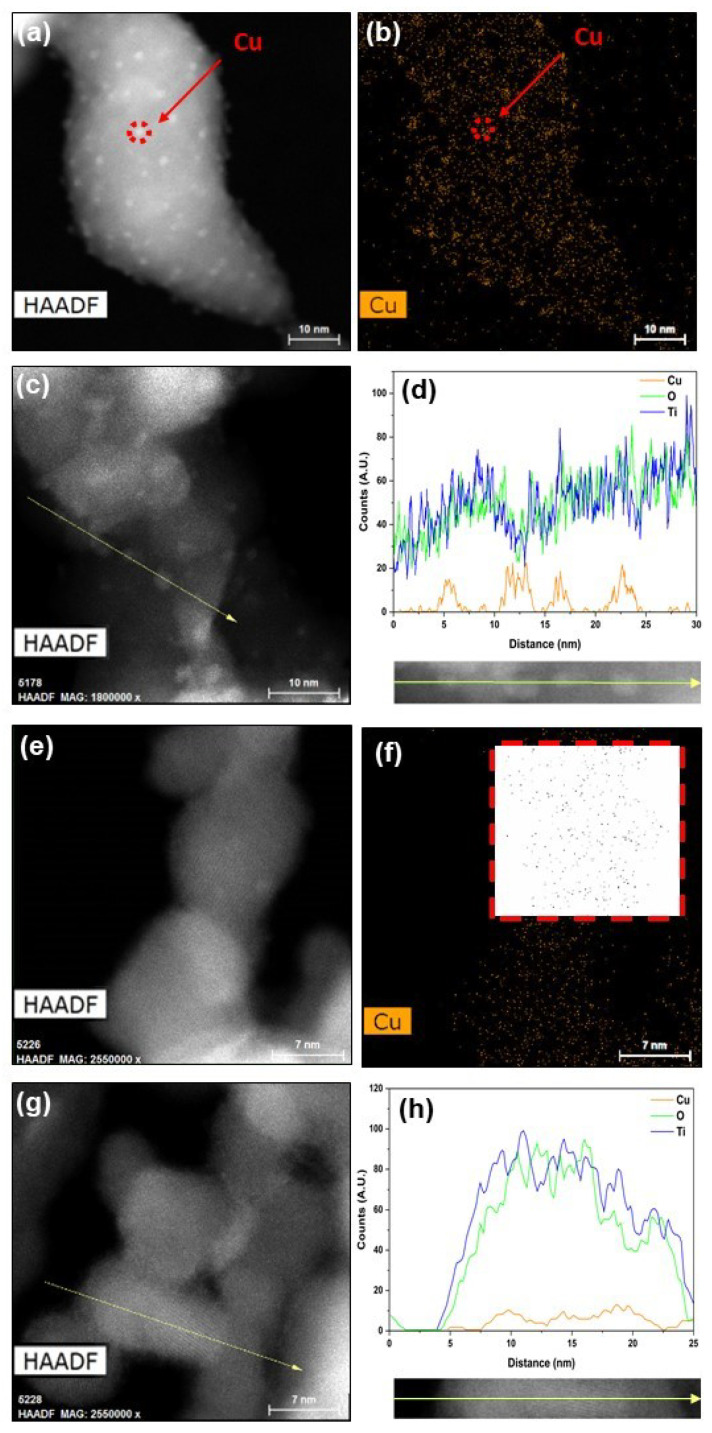
STEM-HAADF images and corresponding Cu elemental maps of Cu/TiO_2_-Ar (**a**,**b**) and Cu/TiO_2_-He (**e**,**f**) samples; STEM-HAADF images and associated EDS line scan analysis recorded along the yellow arrow performed on Cu/TiO_2_-Ar (**c**,**d**) and Cu/TiO_2_-He (**g**,**h**). Inset in (**f**) highlights the visualisation of Cu element (black spots on white background) by means of a treatment operated on ImageJ software.

**Figure 3 nanomaterials-13-00792-f003:**
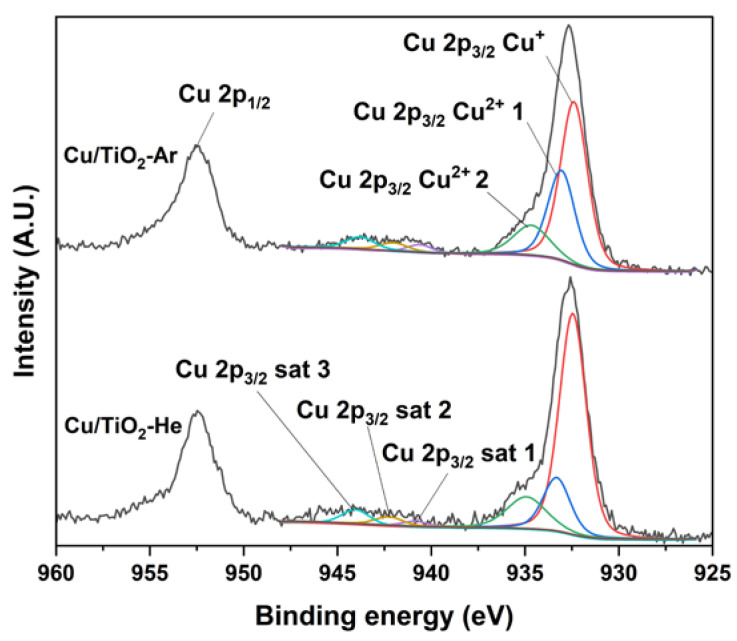
Cu 2p core-level spectra comparison of Cu/TiO_2_-He and Cu/TiO_2_-Ar samples.

**Figure 4 nanomaterials-13-00792-f004:**
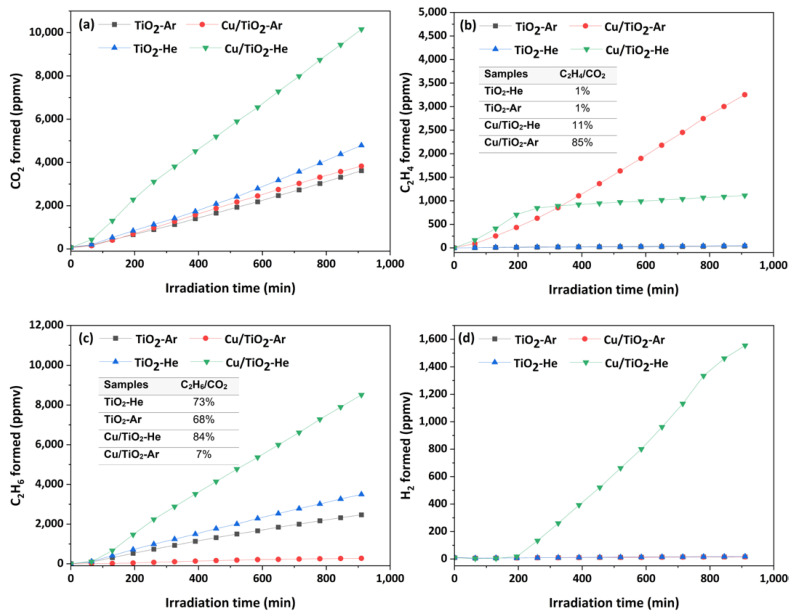
Major products formed from the photo-decarboxylation of PA (1 vol%) under UVA light: CO_2_ (**a**), C_2_H_4_ (**b**), C_2_H_6_ (**c**), H_2_ (**d**). Ratios are calculated at 910 min of irradiation (see also Appendix A).

**Table 1 nanomaterials-13-00792-t001:** Characteristics of TiO_2_ and Cu_x_O_y_/TiO_2_ elaborated under He and Ar atmospheres. (Unless specified, characterisation results concern annealed powders).

Samples	Production Rate (g/h)	C Content before Annealing (wt%)	Mass Loss after Annealing (%)	BET Surface Area (m².g^−1^) (Diameter ^a^ (nm))	TEM Diameter ^b^ (nm)	Crystallinity (Phase Fraction)		Cu Content (wt%)
						Anatase	Rutile	
TiO_2_-He	5.1	34.5	41	111 (13.6)	15 ± 4	87%	13%	-
TiO_2_-Ar	10.3	44.7	49	81 (18.6)	16 ± 6	76%	24%	-
Cu/TiO_2_-He	4.5	17.9	22	138 (10.8)	12 ± 3	89%	11%	1.76 ± 0.04
Cu/TiO_2_-Ar	9.5	53.9	58	40 (35.7)	31 ± 11	53%	47%	1.91 ± 0.04

^a^: Calculated from 6000/specific surface area (Brunauer–Emmett–Teller, BET) [m^2^.g^−1^] × density [g/cm^3^]) formula. Density takes into account anatase and rutile proportions determined using XRD as well as Cu content measured via ICP-OES. ^b^: Diameter estimated from the measure of 100 particles using ImageJ 1.53t software.

**Table 2 nanomaterials-13-00792-t002:** XPS analysis of TiO_2_ and Cu_x_O_y_/TiO_2_ elaborated under He and Ar atmospheres.

Samples	Ti 2p_3/2_ (eV) [18]		Cu 2p_3/2_ (eV) [18,21]		Cu^2+^ (at%)	Cu/Ti
	BE	FWHM	Cu^0^ + Cu^+^	Cu^2+^		
TiO_2_-He	458.5	1.1	-	-	-	-
Cu/TiO_2_-He	458.5	1.3	932.5	933.3; 934.9	36.7	0.06
TiO_2_-Ar	458.5	1.1	-	-	-	-
Cu/TiO_2_-Ar	458.5	1.5	932.4	933.1; 934.7	49.3	0.14

## Data Availability

The data that support the findings of this study are available from the authors on reasonable request.

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
