# Peer review of "One-Step Synthesis of CuxOy/TiO2 Photocatalysts by Laser Pyrolysis for Selective Ethylene Production from Propionic Acid Degradation"

_nanomaterials, 2023, doi:10.3390/nano13050792_

Round 1

Reviewer 1 Report

In this manuscript, Juliette Karpiel et.al. prepared CuOx/TiO2 composites for ethylene production from propionic acid degradation. Herein, the effect of synthesis atmosphere was systematically investigated. There are some minor issues to be well addressed before the publication approval in this journal.

1) As for the operation mechanism, homojunctions also existed between rutile and anatase TiO2. Small lattice mismatch and more interfaces are beneficial for more efficient electron transfer during the photocatalytic process. See the references-ACS Appl. Mater. Interfaces, 2021, 13, 47, 56485-56497; Adv. Mater. 2021, 33, 2005303. The authors should add the discussion in this section.

2) How to confirm the optimal componential combination of the photocatalysts? The optimization process was absent.

Reviewer 2 Report

This paper revealed that the laser pyrolysis in Ar atmosphere makes more effective photocatalytic materials than in He atmosphere.

The differences in properties of the produced photocatalytic materials were studied between Ar and He atmosphere.

The comments are as follows.

1.       Sections 3.2 and conclusion.

The reason why such kind of nanoparticles differences are generated by the atmosphere difference should be discussed more.

Please add the discussion.

2. lines 73-74.

Write the wave length of the CO2 laser.

3. line 80.

Write the reactor size.

4. line 83.

Add the laser beam shape and laser diameter. Was the laser beam focused or not?

5. line 102.

Explain BET method briefly. What kinds of physical properties are characterized? How is the principle of detection?

6. line 114.

At the abbreviation, the paper is missing the word “plasma”.

7. line139-140.

Describe “the cooling effect and less efficient confinement” in more detail.

8. line 152 , 160.

Do not use abbreviation for “BET” here.

9. line166-167.

Please explain what resulted in the higher temperature.

10. line 170.

Write “anneal” condition.

11. line 183-184.

Why is it more efficient?

12. line 219.

What is “HAADF”?

13. line 243.

Do not use abbreviation for “PA” here.

14. line 255.

In Figures right up shows only three data lines and in Figure right down shows two data lines.

Explain why some lines are missing.

15. line 255.

Do not use abbreviation for “PA” and “UVA”.

16. line 297.

Change “NPs” to nanoparticles.

END

Reviewer 3 Report

The manuscript is devoted to the processing of the production of ethylene (C2H4) (one of the key molecules in the chemical industry) using photocatalysis of carboxylic acids that can be produced from the fermentation of organic wastes. It is shown that photocatalysis is extremely efficient when using CuxOy/TiO2 nanoparticles under UVA light particularly if Cu-based nanoparticles are bonded on TiO2 in the form of 2 nanometer-sized clusters. In manuscript is described the way of production of such nanoparticles in one step process using laser pyrolysis in an inert atmosphere that can be applied on an industrial scale. Their superior efficiency and selectivity in photocatalysis were demonstrated in the production of ethylene from propionic acid and compared with other materials.

The manuscript is interesting from a scientific and applied point of view, it is well-organized, the experiment is described in detail, discussion of results sounds convincing. My suggestion is published as it is.

Reviewer 4 Report

Karpiel et. al. synthesis Cu/TiO2 photocatalysts by a laser assisted pyrolysis method. They compare Ar and He atmosphere during synthesis. They test the materials in photocatalytic degradation of propanionic acid and observe the products. They find larger Cu particles lead to a higher selectivity for ethylene. Considering the high scale of synthesis, the results are very good. The paper is suitable for publication in this journal. 

I have some minor comments:

1)Please include more details on the GC:

- what is the carrier gas?

- How seperate are the peak in the chromatogram? the authors should show an example and explain their confidence in assigning the peaks.

2) How much propanionic acid is converted after 900 min? How many (or what %) unaccounted-for products are there? It is common to show the % of each product in a stacked bar chart (sometimes called percentage bar chart).

Reviewer 5 Report

The paper nanomaterials-2211518 describing ethylene production from propionic acid degradation is written very consistently with reasons why this reaction is important from a practical point of view. The presentation of the results shows the effectiveness of using CuxOy/TiO2. The manuscript could be recommended for publication in its present form, however, the curiosity of the reader arouses the desire to compare the efficiency of Cu with other metals analyzed earlier (Au, Pt, Pd). A brief comparison of the results for different metals would underline the importance of this work.

Other less significant remarks concern the addition of a reference in the sentence “Nonetheless, very few research articles describe the photocatalytic production of ethylene.” (p1.39) and an explanation of the meaning P25 (p2, 48).

Round 2

Reviewer 2 Report

The revised manuscript was properly corrected..